# Cross-tissue coordination between SLC nucleoside transporters regulates reproduction in *Caenorhabditis elegans*

Youchen Guan[1,2], Yong Yu[3], Shihong M. Gao[1], Lang Ding[1,4], Qian Zhao[1], Meng C. Wang[1]*

1 Janelia Research Campus, Howard Hughes Medical Institute, Ashburn Virginia, United States of America, 2 Molecular and Cellular Biology Graduate Program, Baylor College of Medicine, Houston Texas, United States of America, 3 Huffington Center on Aging, Baylor College of Medicine, Houston Texas, United States of America, 4 Chemical, Physical & Structural Biology Graduate Program, Baylor College of Medicine, Houston Texas, United States of America

¤Current address: State Key Laboratory of Cellular Stress Biology, School of Life Sciences, Faculty of Medicine and Life Sciences, Xiamen University, Xiamen, China

* mengwang@janelia.hhmi.org

## Abstract

Metabolism is fundamental to organism physiology and pathology. From the intricate network of metabolic reactions, diverse chemical molecules, collectively termed metabolites, are produced. In multicellular organisms, metabolite communication between different tissues is vital for maintaining homeostasis and adaptation. However, the molecular mechanisms mediating these metabolite communications remain poorly understood. Here, we focus on nucleosides and nucleotides, essential metabolites involved in multiple cellular processes, and report the pivotal role of the SLC29A family of transporters in mediating nucleoside coordination between the soma and the germline. Through genetic analysis, we discovered that two *Caenorhabditis elegans* homologs of SLC29A transporters, Equilibrative Nucleoside Transporter ENT-1 and ENT-2, act in the germline and the intestine, respectively, to regulate reproduction. Their knockdown synergistically results in sterility. Further single-cell transcriptomic and targeted metabolomic profiling revealed that the ENT double knockdown specifically affects genes in the purine biosynthesis pathway and reduces the ratio of guanosine to adenosine levels. Importantly, guanosine supplementation into the body cavity/pseudocoelom through microinjection rescued the sterility caused by the ENT double knockdown, whereas adenosine microinjection had no effect. Together, these studies support guanosine as a rate-limiting factor in the control of reproduction, uncover the previously unknown nucleoside/nucleotide communication between the soma and the germline essential for reproductive success, and highlight the significance of SLC-mediated cell-nonautonomous metabolite coordination in regulating organism physiology.

**Data availability statement:** The snRNA-seq data that support the findings of this study are publicly available from GEO repository with the identifier: GSE265917.

**Funding:** M.C.W. receives current support from Howard Hughes Medical Institute. S. M. G., Q. Z., and M.C.W. received salary from HHMI. HHMI had no role in study design, data collection and analysis, decision to publish, or preparation of the manuscript.

**Competing interests:** The authors have declared that no competing interests exist.

## Author summary

Metabolism is essential for life, involving a complex network of chemical reactions that requires a well-organized system to maintain efficiency. This includes the optimal allocation of resources and the dynamic exchange of metabolic products between various compartments within an organism. Solute carriers (SLCs) are the largest family of transporters for metabolic products across the animal kingdom. In our research, we investigated how specific SLC transporters collaborate to move key metabolic products between different tissues. We identified two SLC transporters, Equilibrative Nucleoside Transporter ENT-1 and ENT-2, which are vital for transporting guanosine, a purine nucleoside, to support successful reproduction in the nematode *Caenorhabditis elegans*. We discovered that ENT-2 acts in the gut to export guanosine to the surrounding body cavity, while ENT-1 functions in the germline to import guanosine from the body cavity. When both transporters are disrupted, the animals experience significant reproductive defects. Our study underscores the importance of coordinated activity between SLC transporters in different tissues to maintain organism health. A breakdown in this communication can result in metabolic imbalances and physiological dysfunction.

## Introduction

In multicellular eukaryotic organisms, a complex network of small metabolites, encompassing amino acids, glucose, lipids, and nucleotides/nucleosides, serves as the foundational framework for cellular homeostasis. Perturbations in this intricate metabolic network have been linked to various chronic diseases such as diabetes, cancers, and neurodegenerative disorders [1–3]. In particular, nucleosides and nucleotides are essential for DNA replication and RNA synthesis to enable cell growth and division. Imbalances in nucleotide species can disrupt genome stability, mitochondrial activity, cell proliferation, muscle integrity, germline maintenance, and organism development [4–7].

The synthesis of DNA and RNA precursors, nucleotides, primarily involves *de novo* and salvage pathways. In the *de novo* pathway, nucleotides are synthesized from various substrates such as amino acids, PPRP (Phosphoribosyl pyrophosphate), and tetrahydrofolate [8], through a series of enzymatic steps. On the other hand, the salvage pathway, known for its higher energy efficiency, directly produces nucleotides from free purine/pyrimidine nucleobases and nucleosides. While nucleobases and nucleosides are preferentially salvaged within the cell, their uptake from the extracellular space also plays a crucial role in regulating nucleotide balance, especially in tissues with high demand. In *C. elegans,* the germline, which carries cells undergoing rapid proliferation and division, requires a large amount of nucleotide synthesis. We recently found that the mitochondrial GTP but not ATP level in the germline regulates reproductive activities during aging, which is coupled with bacterial inputs from the intestine [9].

Solute carrier (SLC) transporters are one of the two major transporter superfamilies responsible for the transport of a diverse range of small molecules across the plasma membrane and subcellular organelle membranes. These transporters are essential for cellular homeostasis, regulating the uptake and efflux of various vital nutrients such as glucose, amino acids, fatty acids, vitamins, and ions [10]. They also play a significant role in whole-body physiology, as many are expressed in a tissue-specific manner [11]. Equilibrative Nucleoside Transporter (ENT) family are encoded by SLC29 family genes, and mediate the sodium independent transportation of different nucleobases and nucleosides across membranes [12,13]. Despite the relatively well-studied role of ENT transporters in the cell-autonomous regulation of nucleosides, understanding whether and how they coordinate to mediate nucleoside transport across tissues remains limited. In this work, we found that *C. elegans* ENT-1 and ENT-2 transporters control nucleoside transport between the soma and the germline. Specifically, ENT-2 transports nucleosides from the intestine to the body cavity/pseudocoelom, while ENT-1 facilitates their transportation from the pseudocoelom to the germline. Their synergistic action is essential for reproduction. Moreover, our studies suggest the substrate specificity of ENT-1/2 towards guanosine, through the integration of single-cell transcriptomic profiling, targeted metabolomic profiling, and chemical screening. These findings serve as an example of SLC transporters functioning in different tissues to coordinate the cell non-autonomous communication of vital metabolites, thereby supporting organism physiology.

## Results

### ENT-1 and ENT-2 work together to regulate reproduction

In humans, four ENT members exhibit varying tissue abundance and subcellular distribution [12]. *C. elegans* possesses seven ENT family members: ENT-1 to ENT-7. Despite significant separation in the phylogenetic tree and limited amino acid sequence similarity between *C. elegans* ENTs and human ENTs (Fig 1A), the predicted protein structures of ceENTs based on AlphaFold2 are highly conserved with human ENT1 (Fig 1B) [14]. Among *C. elegans* ENTs, ENT-6 appears relatively divergent from others, while ENT-4, ENT-5, and ENT-7 are relatively close to each other (Fig 1A). ENT-1 and ENT-2 exhibit a notably close relationship (Fig 1A) and share 84% identity in coding exonic nucleotide sequences and 94% in amino acid sequences [15], which has been previously suggested to result from a relatively recent gene duplication event [16].

In *C. elegans* adulthood, only germline cells continue to undergo proliferation and division. To investigate whether successful reproduction depends on nucleoside transport from other tissues into the germline, we examined reproduction in worms where ENTs were knocked down by RNA interference (RNAi). We found that RNAi knockdown of either *ent-1* or *ent-2* but not other *ents* reduces the total progeny number, by 9.8% and 19.8%, respectively. (Fig 1C–1E). RNAi knockdown of either *ent-1* or *ent-2* reduced the number of progenies in the first two days of the reproductive period, and delayed the rate of egg-laying, leading to an extension in the duration of the reproductive process (reproductive lifespan) (Fig 1F and 1G).

Our previous studies revealed that *C. elegans* exhibits different reproductive strategies when exposed to different types of bacteria [9,17]. Specifically, wild-type worms grown on OP50 *E. coli* have extended reproductive lifespan, in comparison with those grown on HT115 *E. coli,* which is commonly used for RNAi. We thus examined the effect of *ent-1* and *ent-2* RNAi knockdown on reproduction in the background of OP50 *E. coli*. We observed a similar extension of reproductive lifespan, however, the reduction in the total progeny number of *ent-1* OP50 RNAi did not reach significance (S1A and S1B Fig).

Next, to confirm the result of RNAi inactivation, we utilized the CRISPR-Cas9 technique [18] to generate knockout mutant strains for both *ent-1* (*ent-1^{KO}*) and *ent-2* (*ent-2^{KO}*) (S1C Fig). We found that in the background of either OP50 or HT115 *E. coli*, the reproductive pattern of the *ent-1^{KO}* mutant is indistinguishable from WT, while the *ent-2^{KO}* mutant slightly increased the progeny number on the third day of the reproductive period (S1D–S1G Figs). Further RT-qPCR analysis revealed elevated *ent-2* mRNA levels in the *ent-1* KO mutant, and increased *ent-1* mRNA levels in the *ent-2* KO mutant (S1H and S1I Fig), which suggests that the loss of one transporter leads to the compensatory induction of the other one. In supporting this idea, we found that the double mutant of *ent-1^{KO}* and *ent-2^{KO}* exhibited complete sterility with

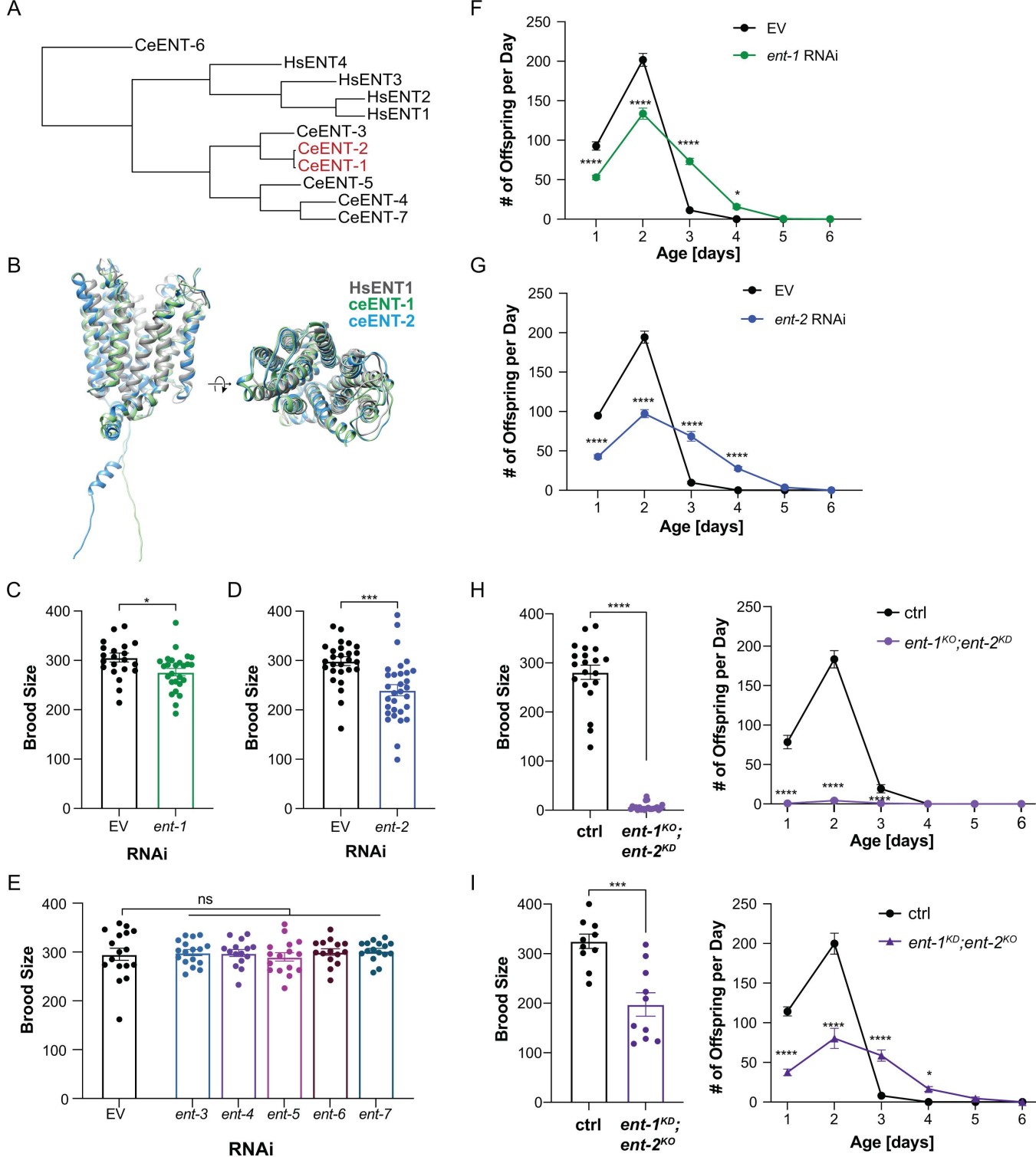

**Fig 1. Specific ENT transporters regulate reproduction.** (A) Phylogenetic analyses of human and *C. elegans* ENTs using protein sequences obtained from Uniprot. (B) Comparison between human ENT1 structure [14], and *C. elegans* ENT-1 (ceENT-1) and ENT-2 (ceENT-1) structures predicted by AlphaFold2. (C) RNA inference (RNAi) knockdown of *ent-1* reduces brood size by 9.8% compared to the empty vector (EV) control. *n* = 22 (EV), n = 26 (*ent-1* RNAi). (D) RNAi knockdown of *ent-2* reduces brood size by 19.8% compared to the EV control. *n* = 28 (EV), n = 31(*ent-2* RNAi). (E) RNAi knockdown of *ent-3*, *ent-4*, *ent-5*, *ent-6*, or *ent-7* does not affect brood size compared to the EV control. *n* = 17 (EV), n = 17 (*ent-3* RNAi), n = 15 (*ent-4* RNAi),

n = 16 (*ent-5* RNAi), n = 15 (*ent-6* RNAi), n = 16 (*ent-7* RNAi). **(F)** RNAi knockdown of *ent-1* alters the daily progeny number compared to the EV control. n = 22 (EV), n = 26 (*ent-1* RNAi). **(G)** RNAi knockdown of *ent-2* alters the daily progeny number compared to the EV control. n = 28 (EV), n = 31(*ent-2* RNAi). **(H)** The *ent-1* knockout (*ent-1^KO*) mutant with *ent-2* RNAi knockdown (*ent-2^KD*) largely reduces the brood size and daily progeny number, compared to the control. n = 20 (ctrl), n = 22 (*ent-1^KO; ent-2^KD*). **(I)** The *ent-2* knockout (*ent-2^KO*) mutant with *ent-1* RNAi knockdown (*ent-1^KD*) reduces the brood size and daily progeny number, compared to the control. n = 10 (ctrl), n = 10 (*ent-2^KO; ent1^KD*). RNAi condition: HT115 *E. coli* (**C-G**); Statistics: *$p < 0.05$; **$p < 0.01$, ***$p < 0.001$, ****$p < 0.0001$, ns: $p > 0.05$. Student's t-test (unpaired, two-tailed) was applied for **C-D**, and the brood size graphs in **H, I**. One-way ANOVA with Holm–Sidak correction for **E**. Two-way ANOVA with Bonferroni's post hoc test for **F-G**, and the daily progeny graphs **H, I**. Data shown as mean ± S.E.M., and performed with three independent biological replicates in **C, D, F, G** and **H**, and two independent biological replicates in **E** and **I**. The "n" values indicate the total number of animals across all replicates.

developmental delay and vulva protrusion, which is consistent with previous observations [15]. We also knocked down *ent-2* by RNAi in the *ent-1* knockout mutant (*ent-1^KO;ent-2^KD*) or knocked down *ent-1* by RNAi in the *ent-2* knockout mutant (*ent-1^KO;ent-2^KD*), referred to as double loss-of-function models. We found that under both conditions, the brood size is largely reduced (Fig 1H and 1I). The phenotype associated with *ent-1^KO;ent-2^KD* is much stronger, close to being sterile (Figs 1H, S2A). These results suggest that ENT-1 and ENT-2 transporters function together to regulate reproduction.

In addition, we measured brood sizes in the loss-of-function mutants of *ent-3 (ok945)*, *ent-4(ok2161)*, *ent-6 (syb7180)*, and *ent-7 (ok1238)* (*ent-5* mutant is currently unavailable). We found that the *ent-4* mutant has a reduced brood size compared to WT (S1J Fig). Interestingly, prior research revealed that ENT-4 is specifically localized at the apical membrane of intestinal cells [19], indicating its potential role in mediating the intestinal uptake of nucleosides from the diet.

## Germline ENT-1 and intestinal ENT-2 coordinate to regulate reproduction

To identify the functional tissues where these two ENTs act to regulate reproduction, we first examined their expression patterns. We generated two CRISPR knock-in lines in which endogenous ENT-1 and ENT-2 are tagged with mNeonGreen and wrmscarlet at the C-terminus, respectively (S2B Fig and S2C Fig). Using these lines, we revealed that ENT-1 is expressed in the germline and intestine, and ENT-2 is expressed in gonadal sheath cells and the intestine (Figs 2A, 2B and S2D, S2E Figs). Furthermore, we examined whether the loss of one transporter leads to the compensatory induction of the other at the protein level by generating and imaging *ent-1::mNeonGreen; ent-2^KO* and *ent-2::wrmScarlet; ent-1^KO* strains. The results showed that ENT-1 protein levels are upregulated in both the intestine and germline in the *ent-2^KO* mutant, and ENT-2 levels are increased in both the intestine and somatic gonad in the *ent-1^KO* mutant (S2F Fig and S2G Fig).

Next, to investigate the tissue-specific roles of *ent-1* and *ent-2* in their coordinative regulation of reproduction, we restored the expression of either *ent-1* or *ent-2* specifically in the intestine of the *ent-1^KO* or *ent-2^KO* mutant, and then performed RNAi knock down of either *ent-2* or *ent-1*, respectively. We found that the intestine-specific restoration of *ent-2*, but not *ent-1,* increases the brood size in the worms with both *ent-1* and *ent-2* knockdown (Fig 2C, 2D). In parallel, we restored the expression of *ent-2* specifically in gonadal sheath cells of the *ent-2^KO* mutant and knocked down *ent-1* by RNAi. We found this restoration does not rescue the reduction in the brood size (Fig 2E). We also attempted to restore the expression of *ent-1* in the germline of the *ent-1^KO* mutant but failed to obtain a stable line.

We thus applied an auxin-inducible degradation (AID) system [20] to deplete ENT-1 proteins selectively in the germline. The AID system utilized a plant-specific F-box protein that recognizes degron-tagged substrates upon auxin binding and induces their degradation by the proteasome (Fig 2F). We generated a CRISPR knock-in line in which endogenous ENT-1 is tagged with degron and V5 tag at the N-terminus (S2H Fig), and subsequently crossed this line with TIR1 tissue-specific expressing lines, germline (*gld-1p::TIR1::mRuby*) and intestine (*ges-1p::TIR1::mRuby*) (Fig 2F). In these crossed lines, the auxin treatment resulted in selective degradation of ENT-1 in the germline or intestine (S2J Fig). With RNAi knockdown of *ent-2,* we observed that auxin-induced germline-specific depletion of ENT-1 resulted in a 2-fold reduction in brood

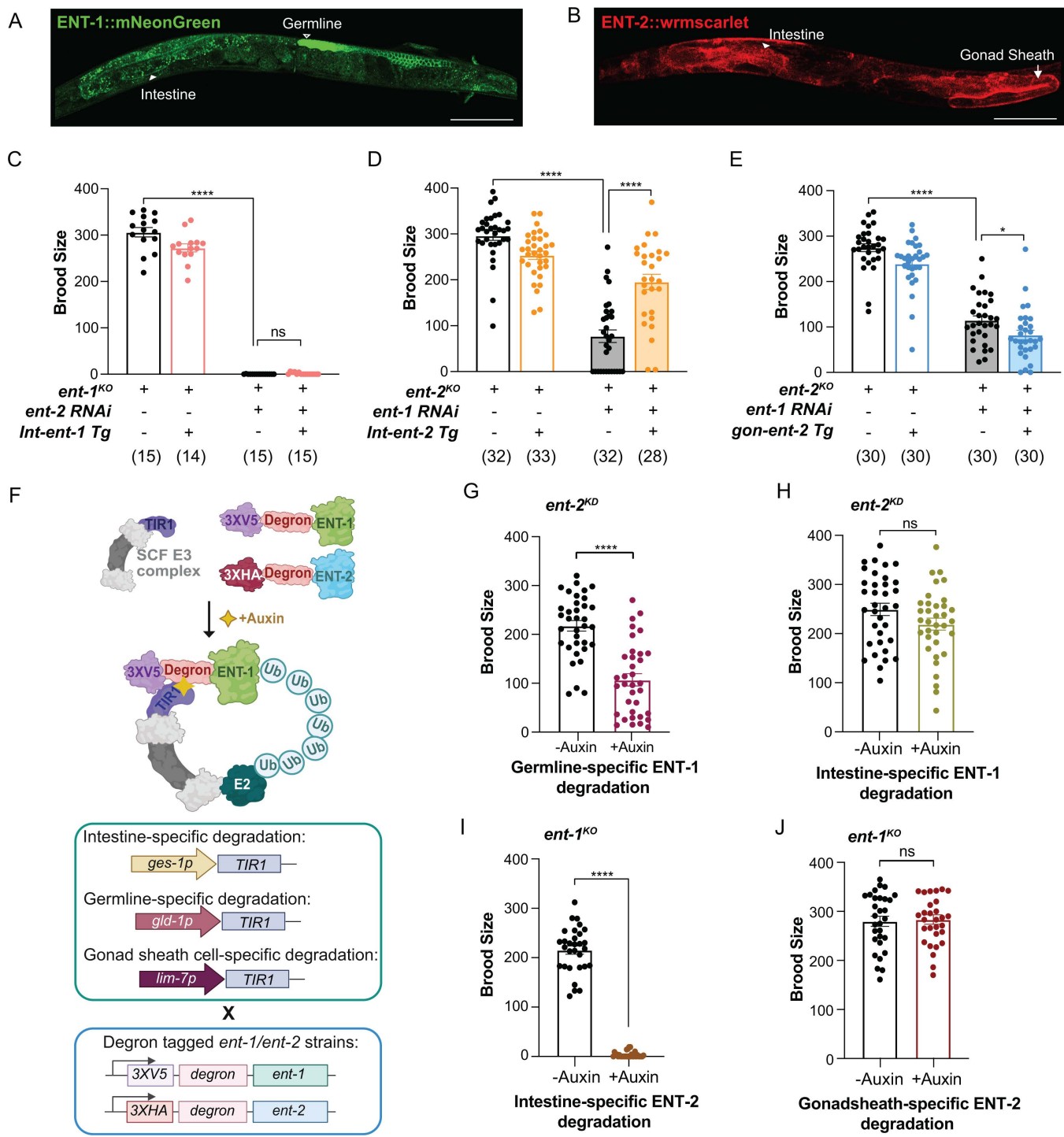

**Fig 2. Germline ENT-1 and intestinal ENT-2 cooperate in regulating reproduction. (A) (B)** The expression patterns of endogenously labeled ENT-1::mNeonGreen and ENT-2::wrmscarlet visualized in their CRISPR knock-in strains. Scale bar: 100 μm. White arrowheads indicate intestine; hollow white arrowhead indicates germline, and white arrows designate gonadal sheath cells. **(C)** Intestine-specific restoration of *ent-1* expression does not rescue the reduced brood size in the *ent-1^KO* mutant with *ent-2* RNAi knockdown. **(D)** Intestine-specific restoration of *ent-2* expression rescues the reduced brood size in the *ent-2^KO* mutant with *ent-1* RNAi knockdown. **(E)** Restoration of *ent-2* expression in somatic gonadal sheath cells does not rescue the reduced brood size in the *ent-2^KO* mutant with *ent-1* RNAi knockdown. **(F)** Illustration of auxin-induced degradation of endogenous ENT-1 tagged with 3XV5 tag and degron and ENT-2 tagged with 3XHA tag and degron in the intestine (*ges-1 promoter*), in the germline (*gld-1 promoter*) and

gonadal sheath cell (*lim-7 promoter*). **(G)** Auxin-induced germline-specific degradation of ENT-1 together with *ent-2* RNAi knockdown reduces the brood size. $n = 23$ (-Auxin), $n = 23$ (+Auxin). **(H)** Auxin-induced intestine-specific degradation of ENT-1 together with *ent-2* RNAi knockdown does not reduce the brood size. $n = 23$ (-Auxin), $n = 23$ (+Auxin). **(I)** Auxin-induced intestine-specific degradation of ENT-2 in the *ent-1^KO* mutant decreases the brood size. $n = 30$ (-Auxin), $n = 30$ (+Auxin). **(J)** Auxin-induced intestine-specific degradation of ENT-2 in the *ent-1^KO* mutant does not affect the brood size. $n = 30$ (-Auxin), $n = 30$ (+Auxin). RNAi condition: HT115 *E. coli* (**C**, **D**, **E**, **G**, **H**). Statistic: $*p < 0.05$; $**p < 0.01$, $***p < 0.001$, $****p < 0.0001$, ns $p > 0.05$. Two-way ANOVA with Bonferroni's post hoc test for **C-E**. Student's t-test (unpaired, two-tailed) between the control group and experimental group was applied for the brood size graphs shown in **G, H, I, J**. Data are presented as mean ± S.E.M. from at least three independent biological replicates, each with around 5–10 animals. The "n" values represent the total number of animals across replicates.

size compared to the control without the auxin treatment (Fig 2G). In contrast, the intestine-specific depletion of ENT-1 did not reduce the brood size compared to the control (Fig 2H), which is consistent with the result using the intestine-specific rescuing strain (Fig 2C).

We have also generated the degron-tagged *ent-2* CRISPR knock-in line and crossed it with the *ent-1^KO* mutant (S2I Fig). This strain was further crossed with the *TIR1* tissue-specific expressing lines, gonadal sheath cell (*lim-7p::TIR1::mRuby*) and intestine (*ges-1p::TIR1::mRuby*) (Fig 2F). In these crossed lines, the treatment of auxin induces specific degradation of ENT-2 either in the gonadal sheath cells or intestine (S2K Fig). Consistent with the findings using tissue-specific rescuing strains, we found that the depletion of *ent-2* specifically in the intestine led to a largely reduced brood size (Fig 2I), while the depletion in gonadal sheath cells does not affect brood size (Fig 2J), when compared to the controls. Together, these results demonstrated that ENT-1 and ENT-2 function in the germline and intestine, respectively to regulate reproduction in a coordinated manner.

Notably, tagging AID to the N-terminus of ENT-1 and ENT-2 partially compromised their function, as shown by a mildly reduced brood size (~200 vs. ~300 in WT) in AID-tagged strains under the mutant background of the other *ent* gene without the Auxin treatment (Fig 2G, 2I). This occurred despite we included a linker between the degron sequence and ENT-1/2. We also attempted C-terminal AID tagging for both transporters and observed even greater functional disruption in these strains, characterized by developmental delays and sterility, resembling the phenotypes of the complete double knockout mutant. While previous studies [20] demonstrated that TIR1 expression and auxin treatment generally do not affect viability, fertility, or development, and the degron sequence is relatively small (44 amino acids) as a tag, our findings emphasize the importance of degron-tagging site selection for transmembrane transporters *in vivo*.

## snRNA-seq profiling reveals the effect of ENTs on purine metabolism

Following our identification of the synergistic effect between ENT-1 and ENT-2 and their tissue-specificity, we next sought to understand how they coordinate the intestine and the germline to regulate reproduction. To this end, we conducted single-nucleus RNA sequencing (snRNA-seq) analysis and profiled cell type-specific transcriptomic changes in the *ent-1^KO*, *ent-2^KO*, and *ent-1^KO;ent-2^KD* worms. Approximately 3,000 worms were collected for each condition, and nuclei were isolated using fluorescence-activated cell sorting (FACS) and profiled using the 10X Genomics platform [21]. Following pre-processing, cell filtering, and quality control, we obtained 56,992 single-nucleus transcriptome profiles, which were subsequently annotated into 14 distinct cell types (Fig 3A). Analysis of cell composition across these conditions revealed a notable decrease in the proportion of germline cells and a higher proportion of somatic cells collected from the *ent-1^KO;ent-2^KD* worms (Fig 3B), which is consistent with the reduced brood size.

We further conducted germ cell pseudotime inference analysis [21], to examine changes in the germline composition. This analysis allowed us to construct a pseudotemporal order of germ cells (Fig 3C), where the x-axis signifies pseudotime denoting the progression from germline stem cells (GSCs) through mitotic cells and meiotic cells to mature oocytes. We found that the GSC number is increased in the *ent-1^KO* single mutant but decreased in the *ent-2^KO* single mutant (Fig 3C), suggesting distinctive alterations in germline homeostasis despite the absence of brood size changes in these single mutants. Moreover, in the *ent-1^KO;ent-2^KD* worms, we observed an increased proportion of germ cells in proliferation

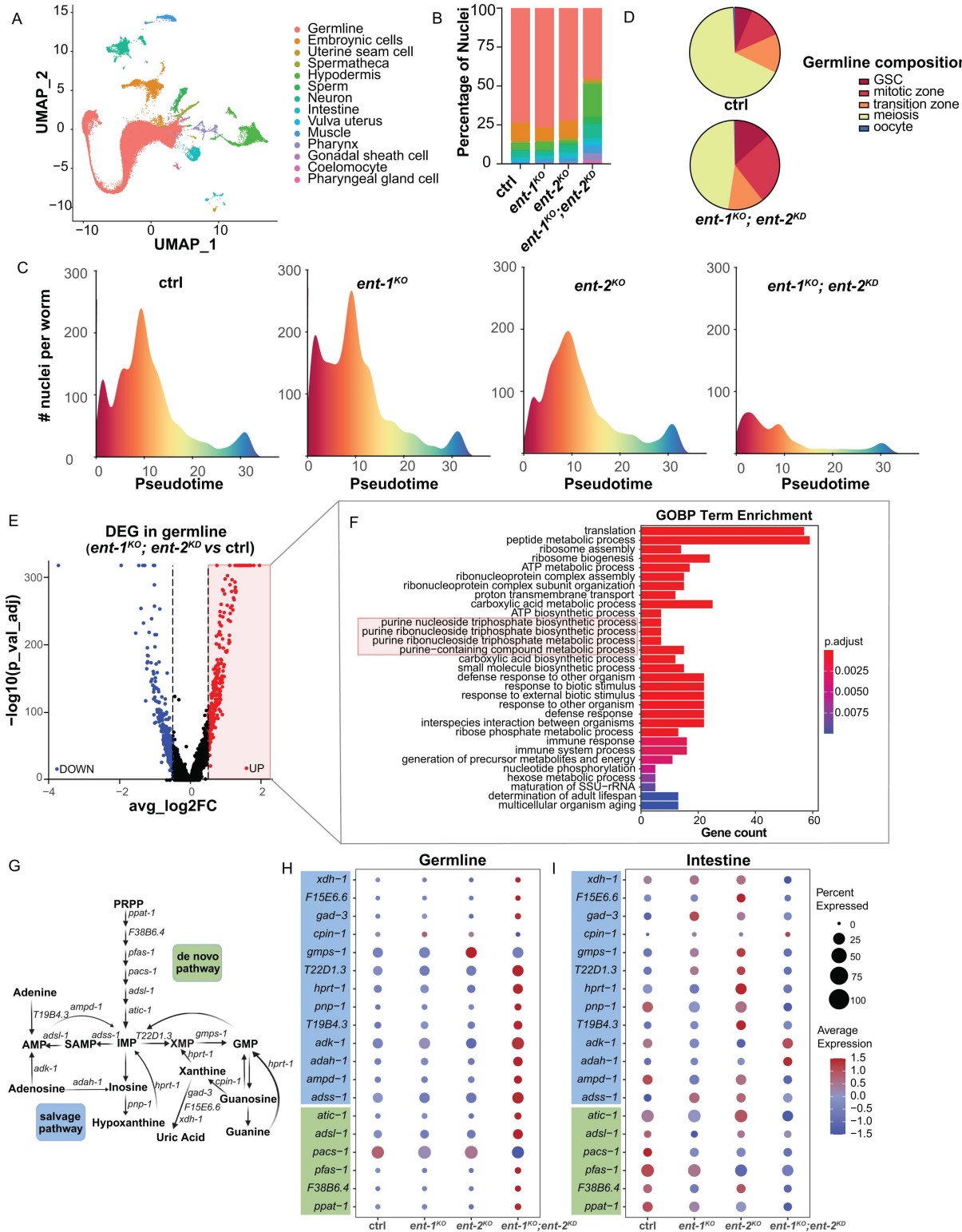

**Fig 3. SnRNA-seq profiling reveals the regulation of purine metabolism by ENT-1/2. (A)** The Uniform Manifold Approximation and Projection (UMAP) plot displays fourteen cell types in ctrl, *ent-1^KO^*, *ent-2^KO^*, and *ent-1^KO^; ent-2^KD^* worms. **(B)** The percentage of different cell types in total captured cells shows decreased germline nuclei in *ent-1^KO^; ent-2^KD^* worms. **(C)** Pseudotime density plots illustrate the distribution of germ cell nuclei across

different pseudotime points for control, *ent-1^{KO}*, *ent-2^{KO}*, and *ent-1^{KO}; ent-2^{KD}* worms. **(D)** The proportion of nuclei within various regions of the germline among total germline nuclei is compared between control and *ent-1^{KO}; ent-2^{KD}* worms. **(E)** The Volcano plot shows differentially expressed genes in the germline when comparing *ent-1^{KO}; ent-2^{KD}* worms vs. their controls. **(F)** Gene ontology (GO) enrichment analysis of the up-regulated genes in the germline of *ent-1^{KO}; ent-2^{KD}* worms compared to their controls. The red box highlights GO terms involved in purine metabolism. **(G)** Schematic illustration of *C. elegans* homologous genes involved in purine metabolism, both *de novo* and salvage pathways, based on the previous study [6]. **(H, I)** Dot plots show the relative expression levels of genes associated with purine metabolism in the germline **(H)** and intestine **(I).** The size of the dots indicates the percentage of nuclei expressing the gene, and the color indicates the average expression level.

stages, including GSC and those in the mitotic zone (Fig 3D). In contrast, the proportion of cells in the meiosis stage decreased (Fig 3D), while the transition zone remained unchanged. These results suggest a defect in germ cell differentiation associated with the loss of both ENT-1 and ENT-2.

We also performed the differential expressed gene (DEG) analysis to reveal molecular changes in the germline upon the loss of ENT-1 and ENT-2. 716 DEG genes were identified through the comparison between *ent-1^{KO};ent-2^{KD}* worms and their controls, with a significance cutoff of p adjust value < 0.05 and |log2FoldChange| > 0.5. Among these genes, 302 were upregulated, while 412 were downregulated (Fig 3E). The Gene Ontology (GO) pathway analysis of the downregulated genes in the germline of the *ent-1^{KO};ent-2^{KD}* worms highlighted pathways associated with cell cycle, chromosome organization, and mRNA processing, which is likely resulted from the loss of differentiated germ cells in these worms (S3A Fig). When analyzing the upregulated genes, we found that GO terms related to purine metabolism are overrepresented in the *ent-1^{KO};ent-2^{KD}* germline (Fig 3F).

A series of purine metabolic genes in both *de novo* and salvage recycling pathways are conserved in *C. elegans* [6] (Fig 3G). When analyzing their expression levels using the snRNA-seq data, we found that most purine metabolic genes showed trends of up-regulation in the *ent-1^{KO};ent-2^{KD}* germline (Fig 3H), while genes encoding enzymes involved in the *de novo* pathway showed trends of down-regulation in the intestine (Fig 3I). However, no clear trends of changes were detected in other somatic tissues such as hypodermis and muscle (S3B and S3C Fig). Furthermore, when analyzing conserved metabolic genes in the pyrimidine pathway (S3D Fig), we did not observe an obvious trend of alterations in their expression in either the germline or intestine of the *ent-1^{KO};ent-2^{KD}* worms (S3E and S3F Fig).

Together, these results suggest that the level of purine, rather than pyrimidine, is likely altered in the germline and intestine upon the loss of both ENT-1 and ENT-2, leading to compensatory transcriptional alterations of purine biosynthesis genes.

### ENT-mediated guanosine transport regulates reproduction

To directly assess whether purine and pyrimidine nucleoside levels change in association with the ENT deficiency, we employed targeted metabolomic analysis by liquid chromatography coupled with mass spectrometry (LC/MS). We compared guanosine, adenosine, inosine, and cytidine levels in *ent-1^{KO}*, *ent-2^{KO}*, and *ent-1^{KO};ent-2^{KD}* worms with their controls. The levels of uridine and thymidine are below the detection sensitivity. Interestingly, we observed that the proportion of guanosine is decreased by ~20% and ~30% in the *ent-1^{KO}* and *ent-2^{KO}* single mutants, respectively (Fig 4A). In the *ent-1^{KO};ent-2^{KD}* worms, the decrease is close to be 3-fold (Fig 4A). For the other two purine nucleosides, adenosine and inosine, their proportion exhibited the opposite trend (Fig 4B and 4C). On the other hand, the proportion of cytidine did not show significant alterations in either *ent-1^{KO}*, *ent-2^{KO}*, or *ent-1^{KO};ent-2^{KD}* worms compared to their controls (Fig 4D). It is worth noting that no reduction in the brood size was observed in the *ent-1^{KO}* or *ent-2^{KO}* single mutant, despite the decrease in guanosine. Hence, the germline is tolerant to a certain level of reduction in guanosine levels while sustaining normal reproduction, but exceeding this threshold would result in reproductive defects.

These results support the predominant role of ENT-1/2 in regulating purine, especially guanosine homeostasis. When integrating the tissue-specificity of ENT-1/2, the germline-specific transcriptional up-regulation of purine metabolic genes,

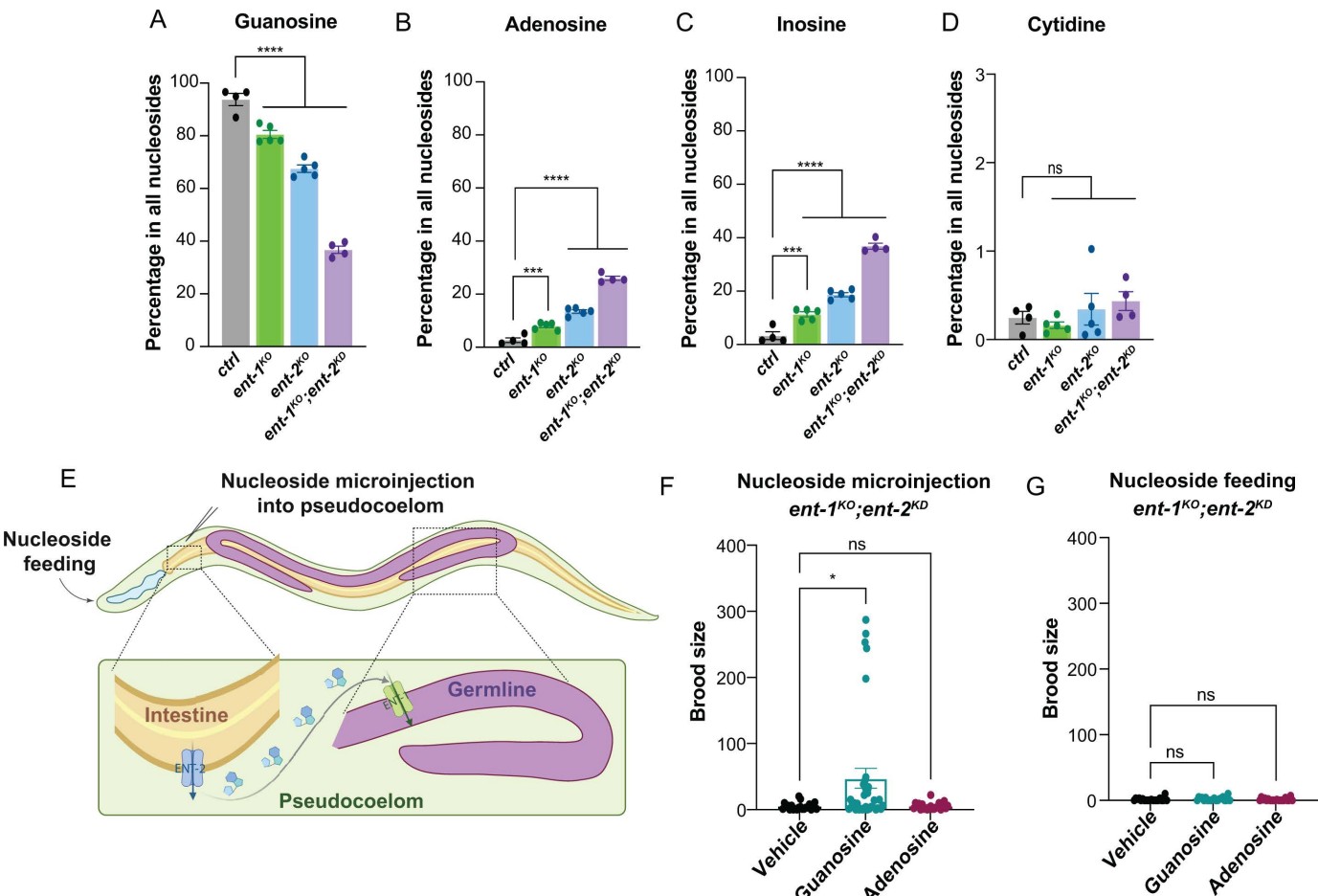

**Fig 4. Guanosine specifically contributes to the regulation of reproduction by ENT-1/2. (A-D)** The percentages of guanosine **(A)**, adenosine **(B)**, inosine **(C)**, and cytidine **(D)** among all nucleosides measured by LC/MS are compared among control, ent-1$^{KO}$, ent-2$^{KO}$, and ent-1$^{KO}$;ent-2$^{KD}$ worms. $n = 4$ (ctrl), $n = 5$ (ent-1$^{KO}$), $n = 5$ (ent-2$^{KO}$), $n = 4$ (ent-1$^{KO}$;ent-2$^{KD}$). **(E)** A proposed summary model illustrating the coordinated regulation of the reproductive process by ENT-2 mediated export of nucleosides from the intestine to the pseudocoelom and ENT-1 mediated uptake of nucleosides from the pseudocoelom to the germline. The model was created via BioRender (https://www.biorender.com). **(F)** Microinjection of guanosine but not adenosine partially rescues the reduced brood sizes of ent-1$^{KO}$;ent-2$^{KD}$ worms. $n = 22$ (DMSO vehicle), $n = 34$ (Guanosine), $n = 22$ (1mM Adenosine), from 3 independent biological replicates. **(G)** Oral supplementation of neither guanosine nor adenosine rescues the reduced brood sizes of ent-1$^{KO}$;ent-2$^{KD}$ worms. $n = 20$ (DMSO vehicle), $n = 20$ (Guanosine), $n = 20$ (Adenosine), from 2 independent biological replicates. Statistic: *$p < 0.05$, ***$p < 0.001$, ****$p < 0.0001$, ns $p > 0.05$. One-way ANOVA with Holm–Sidak correction (**A-D**, and **F-G**). Data shown as mean ± S.E.M.

and the guanosine reduction, we hypothesized that ENT-2 functions in the intestine to export guanosine into the body cavity/pseudocoelom, while ENT-1 is responsible for the uptake of guanosine into the germline (Fig 4E).

In supporting this hypothesis, we confirmed that ENT-2::wrmscarlet predominantly localizes at the basolateral plasma membrane of the intestine (S2E Fig), while ENT-1::mNeonGreen is positioned at the plasma membrane in the germline (S2D Fig). We also injected guanosine or adenosine into the pseudocoelom at the fourth larval (L4) stage to increase their levels in the pseudocoelom. We found that the microinjection of 1mM guanosine can increase the brood size in the ent-1$^{KO}$;ent-2$^{KD}$ worms, but adenosine microinjection showed no effect in restoring reproduction (Fig 4F). However, feeding of either guanosine or adenosine was not able to rescue the brood size in the ent-1$^{KO}$;ent-2$^{KD}$ worms (Fig 4G). We also tried directly injecting nucleosides into the germline of the ent-1$^{KO}$;ent-2$^{KD}$ worms; however, the severe disruption in germline

morphology prevented this attempt. These results highlighted the importance of guanosine levels in the pseudocoelom in the regulation of reproduction.

## Discussion

Our results reveal that the transport of guanosine from somatic tissues to the germline is crucial for sustaining reproduction. ENT-1 and ENT-2 transporters collaboratively maintain guanosine equilibrium among the intestine, pseudocoelom, and germline. The loss of ENT-2 in the intestine decreases guanosine availability in the pseudocoelom, while the loss of ENT-1 in the germline impairs guanosine uptake from the pseudocoelom. In single mutants of either *ent-1* or *ent-2*, the level of guanosine in the germline is reduced yet it remains sufficient to sustain reproduction. However, the simultaneous loss of both ENT-1 and ENT-2 results in a server reduction in germline guanosine levels, ultimately causing sterility.

In hermaphrodite *C. elegans*, the developmental process to generate over 2000 germ cells consumes substantial nucleotides [22], serving as basic building blocks for DNA and RNA synthesis. Equilibrative nucleoside transporters facilitate the transport of nucleosides. Interestingly, in the *ent-1^KO^; ent-2^KD^* double loss-of-function models, we observed transcriptional upregulations of both *de novo* and *salvage* biosynthesis pathways for purine but not for pyrimidine nucleotides. Furthermore, metabolite profiling revealed that guanosine levels are decreased in the single and double *ent* mutants, while levels of adenosine and inosine are increased. In addition, microinjection of guanosine, but not adenosine, partially restored fertility in the double loss-of-function models. Based on these results, we propose that ENT-1 and ENT-2 play critical roles in transporting purine nucleosides, including guanosine, adenosine, and inosine. In the germline to support DNA and RNA synthesis, guanosine transported from the intestine serves as the primary source for producing dGTP/GTP. In contrast, dATP/ATP is less reliant on adenosine but can be derived from alternative sources such as mitochondrial oxidative phosphorylation. In the *ent* single and double loss-of-function mutants, germline DNA and RNA synthesis reduce the guanosine pool, while having little impact on the adenosine/inosine pool. This reduction in germline guanosine levels may trigger a compensatory upregulation of overall purine biosynthesis, which could account for the observed increases in adenosine and inosine levels. However, despite the compensation, guanosine levels remain lower due to its high demand for DNA and RNA synthesis.

Our findings provide new evidence supporting the importance of intestinal nucleoside/nucleotide availability in regulating reproduction. Previous studies have elucidated that intestinal uridine and thymidine levels modulate the expression of the Notch receptor GLP-1 in the germline, thereby controlling *C. elegans* reproduction [4]. In addition, nucleotide imbalance has been demonstrated to signal through the intestinal endonuclease ENDU-2 and CTP synthase CTPS-1 to regulate germ cell proliferation in *C. elegans* [23]. These findings highlight that animals sense pyrimidine nucleotides in the intestine to modulate reproductive activity. Therefore, the availability and balance of pyrimidine and purine nucleoside and nucleotide levels are critical for reproductive success.

ENT-1 and ENT-2 are vital not only for *C. elegans* reproduction but also for development. We observed that CRISPR-generated complete double knockout mutants (*ent-1 KO; ent-2 KO*) exhibit severe developmental delays, developmental arrest, and sterility, making it not feasible to maintain the strain. In this study, we mainly utilized double loss-of-function models, where *ent-1 KO* was combined with *ent-2 RNAi* knockdown, and *ent-2 KO* was combined with *ent-1 RNAi* knockdown. These worms developed to the adult stage but exhibited a sterile phenotype. The relatively milder developmental phenotypes observed in the double loss-of-function models compared to the complete double knockout mutants suggest that reproduction may be more sensitive to guanosine dysregulation than development.

The synergistic effect of ENT-1 and ENT-2 in two different tissues in regulating reproduction is uncommon compared to the synergistic effects observed within the same tissue. No obvious reproductive defects were detected in the single mutants, but the double mutants are sterile. We expect that multiple compensatory mechanisms may help sustain normal reproduction in the single mutants. First, compensation may be mediated through the upregulation of guanosine synthesis via the *de novo* and *salvage* pathways from multiple substrates, such as amino acids. These changes likely occur at the

protein level or through elevating enzymatic activity, which are beyond the resolution of snRNA-seq transcriptomic analysis. Secondly, guanosine in the pseudocoelom can be exported from multiple somatic tissues, but not just the intestine. While our study focused on guanosine transport from the intestine to the germline, other tissues, such as muscle or hypodermis, could also supply guanosine via alternative transporters. In the ENT-2 single mutant, although the guanosine export from the intestine is decreased, ENT-1 can still import some guanosine into the germline from the pseudocoelom. In the ENT-1 single mutant, the guanosine level in the pseudocoelom is not low (maybe even higher), and other transport mechanisms may compensate for the loss of ENT-1 and import some guanosine into the germline to support reproduction, which may involve other SLC transporters, including ENT family members and concentrative nucleoside transporters (CNTs) that leverage ion gradients to actively transport nucleosides against their concentration gradients. It would be interesting for future studies to explore the role of other somatic tissues as well as other transporters in nucleoside transport and metabolism.

In summary, our study underscores the significance of guanosine communication between the soma and germline in the control of reproduction. Members of the human ENT family, such as hENT1 and hENT2, are highly expressed in both the digestive and reproductive systems, as reported in the Human Protein Atlas (proteinatlas.org) [24]. This suggests that cross-tissue coordination of nucleoside transport may similarly play a role in reproduction in more complex organisms. Our work also reveals the pivotal role played by specific SLC transporters in mediating cell non-autonomous metabolite communication. Given the conservation of SLC transporters across species and their broad implications in human diseases, an intriguing future direction would be to investigate mechanisms underlying their cell non-autonomous coordination in mammals.

## Materials and methods

### *C. elegans* strains and maintenance

The strains N2, CA1352 (ieSi64 [*gld-1p::TIR1::mRuby::gld-1 3 UTR + Cbr-unc-119(+)*] II), and CA1209 (ieSi61 [*ges-1p::TIR1::mRuby::unc-54 3'UTR + Cbr-unc-119(+)*] II), and MQD2383 (*hqSi11 [lim-7p::TIR1::mRuby::unc-54 3' UTR + Cbr-unc-119(+)]* II; *daf-2(hq363[daf-2::degron::mNeonGreen]) unc-119(ed3)* III), VC643 *ent-3 (ok945)*, VC1934 *ent-4(ok2161)*, and RB1193 *ent-7 (ok1238)* were obtained from CGC.

Knockout mutants MCW1244 (*ent-1(rax74)* IV) and MCW1245 (*ent-2(rax75)*X) were generated in our lab by using CRISPR/Cas9 technology, as previous outlined by Chen et al., Paix et al. and Arribere et al. [25–27]with modifications. Briefly, a mixture containing tracrRNA (1µg/µl), crRNAs (0.5µg/µl each for one on the 5' and one on the 3' of the target gene), dpy-10 crRNA (0.16µg/µl), and Cas9 protein (0.05µg/µl) was microinjected into the gonads of N2 young adult animals. Each injected worm was then placed on individual plates, and the non-Dpy F1 progenies from the plates which contains animals with Dpy phenotypes were individualized to single plates for further analysis. After confirming the deletion region through PCR and Sanger sequencing of F1 animals, homozygous F2 animals carrying the knockout mutation were selected and individualized. The knockout deletion strains were backcrossed to wild-type N2 for at least four times before running reproductive experiments.

Strains that carry extrachromosomal arrays: MCW1589(*ent-2(rax75)*X; raxEx624[*lim-7p::ent-2cds::sl2RFP::tbb-2 3'UTR*; *lin-44p::GFP*], MCW1634(*ent-1(rax74)* IV; raxEx628[*ges-1p::ent-1cds::sl2RFP::unc-54 3'UTR*; *lin-44p::GFP*], MCW1636(*ent-2(rax75)*X; raxEx630[*ges-1p::ent-2cds::sl2RFP::unc-54 3'UTR*; *lin-44p::GFP*] were generated by microinjecting DNA mixture containing linearized expression construct and co-injection marker lin-44p::GFP into the corresponding gonad of *ent-1* or *ent-2* knockout young adult animals.

PHX4218(*ent-1(syb4218) ent-1::mNeonGreen* IV), PHX5055(*ent-2(syb5055) ent-2::wrmscarlet::3Xflag* X), PHX7376(*ent-1(syb7376) 3XV5::degron::ent-1* IV), PHX7457(*ent-2(syb7457) 3XHA::degron::ent-2* X), and PHX7180 *ent-6 (syb7180)* were generated via CRISPR/Cas9 genome editing by SunyBiotech (Fuzhou, China).

MCW1638 (*syb7376[3XV5::degron::ent-1]*IV; ieSi61 [*ges-1p::TIR1::mRuby::unc-54 3'UTR + Cbr-unc-119(+)*]II) and MCW1639 (*syb7376[3XV5::degron::ent-1]IV*; ieSi64 [*gld-1p::TIR1::mRuby::gld-1 3'UTR + Cbr-unc-119(+)*] II) were generated by crossing PHX7376 with CA1209 or CA1352.

MCW1640 (*syb7457[3XHA::degron::ent-2]*X; ieSi61 [*ges-1p::TIR1::mRuby::unc-54 3'UTR + Cbr-unc-119(+)*] II and MCW1641 (*syb7457[3XHA::degron::ent-2]*X; ent-1(rax74) IV; hqSi11 II[*lim-7p::TIR1::mRuby::unc-54 3' UTR + Cbr-unc-119(+)]* II) were generated by crossing PHX7457 with CA1209 or the strain hqSi11 [*lim-7p::TIR1::mRuby::unc-54 3' UTR + Cbr-unc-119(+)*] II which were obtained from crossing MQD2383 and N2.

MCW1643 (*syb7457[3XHA::degron::ent-2]*X; ent-1(rax74) IV; ieSi61 [*ges-1p::TIR1::mRuby::unc-54 3'UTR + Cbr-unc-119(+)*] II and MCW1644 (*syb7457[3XHA::degron::ent-2]*X; ent-1(rax74) IV; hqSi11 II[*lim-7p::TIR1::mRuby::unc-54 3' UTR + Cbr-unc-119(+)*] II) were generated by crossing MCW1640 or MCW1641 with MCW1244.

All the *C. elegans* strains were grow and maintained non-starved for at least three generations at 20˚C on NGM agar plates seeded with OP50 E.coli using standard protocols [28] before experiments. The E. coli strain HT115 (DE3) and OP50 RNAi strain (OP50 bacteria [*rnc14::DTn10 laczgA::T7pol camFRT*] generated by our lab [29] were used for RNAi experiments.

### RNA interference (RNAi) experiments

RNAi libraries from Dr. Julie Ahringer's lab were utilized in the study [30]. RNAi clones for *ent-1, ent-2, ent-3, ent-5, and ent-6* were obtained from the Ahringer library. RNAi clones for *ent-4* and *ent-7* were created in our lab using L4440 as the vector backbone and *ent-4* or *ent-7* transcript fragments as inserts. For OP50 RNAi experiments, RNAi plasmids were transformed into the genetically modified competent OP50 bacteria [rnc14::DTn10 laczgA::T7pol camFRT], generated by our lab [29]. All RNAi colonies were selected for resistance to both 50 µg ml−1 carbenicillin and 50 µg ml−1 tetracycline. RNAi bacteria were cultured for 14 hours in LB with 25 µg ml−1 carbenicillin, then seeded onto RNAi agar plates containing 1 mM IPTG and 50 µg ml−1 carbenicillin. Each RNAi bacteria clone was allowed to dry on the plates before overnight incubation at room temperature to induce dsRNA expression.

### Molecular cloning of expression construct

All the expression plasmids were generated via the Gibson Assembly (NEB). The *ent-1* and *ent-2* coding sequences were PCR-amplified from *C. elegans* cDNA and then fused together with *sl2-RFP* sequence by fusion PCR. The *ent-1::sl2RFP* and *ent-2::sl2RFP* fragments were then ligated into the tissue-specific promoter vectors.

### Measure the brood size

Synchronized L1 worms obtained from egg preparation were placed onto 6 cm NGM plates, each seed with corresponding bacteria for the experimental condition, and then incubated at 20 °C. Upon reaching the L4 stage, individual worms were transferred to new plates. Subsequently, they were transferred to new plates every day until reproduction ceased. Plates with progenies were stored at 20°C until progeny reached the L4 stage for counting. Total brood size was determined by summing viable progeny produced daily by each worm.

### Structural stimulation by AlphaFold2

AlphaFold2 predictions for the structures of ceENT-1 (UniProt: G5EDJ3) and ceENT-2 (UniProt: Q93871) were retrieved from The AlphaFold Protein Structure Database (https://alphafold.ebi.ac.uk/) [31,32]. Molecular graphics for human ENT1, ceENT-1, and ceENT-2 were generated using UCSF Chimera [33].

### Auxin treatment

The L1 worms were places on the NGM plates seeded with bacteria containing auxin 4mM indole-3-acetic acid (IAA), following the protocol outlined by Zhang et al. [20]. A 400 mM stock solution of IAA in ethanol was prepared, filtered through a 0.22µm filter, and stored at 4°C for up to 1 month. This stock solution was diluted into NGM plates at a ratio of 1:100. Control plates were prepared by diluting ethanol into NGM plates at the same ratio. Fresh bacteria were then seeded onto

the plates and stored at room temperature for 1 day to allow bacterial growth. The plates containing auxin were kept in a dark place to prevent photolysis.

## RT-qPCR

Total RNA was extracted from approximately 3000 age synchronized D1 worms using Trizol homogenization, chloroform phase separation, isopropanol precipitation, and subsequent washing with 75% ethanol. cDNA synthesis utilized the amfiRivert Platinum cDNA Synthesis Master Mix (GenDEPOT), followed by quantitative PCR with the Kapa SYBR fast qPCR kit (Kapa Biosystems) in a 96-well Eppendorf Realplex 4 PCR machine (Eppendorf).

All presented data are from at least six independent biological samples and were normalized to *rpl-32* as an internal control.

## Fluorescent microscopy

*C. elegans* were immobilized in 1% sodium azide in M9 buffer and positioned on a 2% agarose pad between a glass microscopic slide and coverslip for imaging. Endogenous ENT-1::mNeonGreen and ENT-2::wrmscarlet expression patterns were visualized using a Nikon CSU-W1 spinning disk confocal microscopy system equipped with a 20x objective, oil immersion 60x and 100x objective. Degron-mediated protein degradation was assessed using a laser scanning confocal FV3000 (Olympus, US) with an oil immersion 60x objective.

## Injection of nucleoside into the pseudocoelom

Stock solutions of nucleosides are stored at 100mM in DMSO at -20°C for up to 1 month. Fresh 1mM nucleoside injection solutions are prepared before each injection by dissolving the 100mM stock solution in egg buffer [118 mM NaCl, 48 mM KCl, 2 mM MgCl2, 2 mM CaCl2, and 25 mM Hepes (pH 7.3)] at a ratio of 1:100. The control injection solution consists of DMSO dissolved at a ratio of 1:100. Mid-stage L4 animals were injected with either the nucleoside solution or DMSO control solution into the pseudocoelom, with the injection needle inserted into the pharyngeal region. Successful injection is confirmed by the observation of liquid flow in the pseudocoelom.

## Immunostaining

Approximately 200 Adult animals were initially transferred to unseeded NGM plates to minimize bacterial presence. M9 solution containing 0.4μM levamisole was then added to immobilize animals on the plates. Subsequently, these animals, along with the M9 solution containing 0.4μM levamisole, were transferred to a glass dissection plate for dissection. Dissection involved using two 25 gauge syringe needles to extrude the gonad arm and intestine completely. The dissected worms were fixed with 4% PFA for 10 minutes at room temperature in darkness, followed by washing twice with PBST and post-fixation in −20°C methanol for 1 hour. After three washes with PBST, the specimens were then blocked with 5% BSA in PBST for 30 minutes. Following blocking, a 200μL volume of primary antibody solution (in PBST) [anti-HA antibody (1:200): HA-Tag (C29F4) Rabbit mAb #3724; anti-V5 antibody (1:200): V5 Tag Monoclonal Antibody (SV5-Pk1) (R960-25)] was applied and incubated overnight at 4°C. The cut worms were then washed three times with PBST and incubated with secondary antibody [1:500 dilution of Alexa 488 conjugated antibodies from Invitrogen] for 1 hour at room temperature. After three additional washes with PBST, the specimens were resuspended in a glycerol antifade reagent and mounted on agarose pads before imaging.

## Nucleoside dietary supplementation

For dietary nucleoside supplementation, nucleoside powder was dissolved directly to a final concentration of 100mM in standard NGM liquid medium immediately prior to pouring into plates. These plates were then seeded with RNAi E. coli and allowed to grow overnight before use.

### Single-nucleus RNA sequencing analysis

**Nuclei isolation.** We followed the nuclei isolation protocol similar to that described by Gao et al. [21]. Briefly, worms were washed three times with PBS, collected in 1.5 mL tubes, and homogenized in 100 µl of homogenization buffer [34] using a pestle motor on ice. To prevent nuclei adhesion, all equipment was pre-coated with homogenization buffer or PBS. The homogenate was further processed in an autoclaved Dounce tissue grinder, with sequential strokes using loose and tight pestles to ensure thorough disaggregation without foaming. After filtering through cell strainers, the nuclei were pelleted by centrifugation, resuspended in PBS with additives, and prepared for sorting.

**Nuclei sorting.** The nuclei were stained with Hoechst 33342 for DNA content visualization and sorted using a Sony MA800 sorter. Gating of sorting was the same as described by Gao et al. [21]. Post-sorting, nuclei were checked for concentration and morphology before proceeding with 10X Chromium Controller.

**Library preparation and sequencing.** The quality-checked nuclei were encapsulated using a 10X Chromium Controller, and libraries were prepared according to the 10X Chromium Single Cell 3' v2/v3 Solution protocol, selecting appropriate indexing options. Sequencing was performed on a NovaSeq 6000 system, using 26 cycles for Read 1, 8 cycles for the i7 index, and 98 cycles for Read 2, as recommended.

**Single-nucleus RNA-seq data preprocessing.** Raw sequences were processed using Cell Ranger 6.0 (10x Genomics), aligned to the *C. elegans* genome (WS282), and assembled into feature-barcode matrices. Doublet exclusion was performed per 10X Genomics guidelines, utilizing Doubletfinder for accurate detection. Subsequent analyses, including data integration, dimensional reduction, and clustering, were conducted using Seurat 4.0.5 to identify and characterize cell populations.

**Cell type annotation and analysis.** For cell type annotation, we predominantly employed a reference-based mapping approach using the SingleR package [35], which allowed us to automatically match the expression profiles of our cells to those from established reference datasets. This analysis predominantly linked cell clusters to specific tissues, as evidenced by previous single-nucleus RNA sequencing datasets produced by Gao et al. [21]. We further enhanced cell type specificity by employing the FindMarkers function within Seurat to identify distinctive marker genes for each cluster. These markers were then cross-referenced with microscopy-based expression profiles and literature-reported tissue markers to confirm cell type assignments. For enrichment analysis and further validation of our findings, we utilized tools available on Wormbase [36].

In addition to broad classification, we also focused on the germline cells, isolating them into a separate Seurat object. This subset underwent a similar analysis pipeline, where we identified highly variable genes and performed CCA integration followed by dimensional reduction and clustering to determine germline subclusters. This refined analysis enabled us to explore cell development within the germline specifically.

**Germline trajectory analysis.** Utilizing the refined germline cluster data, we applied Slingshot for trajectory analysis, which utilized cluster labels and UMAP embeddings to construct cell lineages and ascertain pseudotime trajectories. This analysis revealed the developmental progression from germline stem cells, showing low pseudotime values, to mature oocytes, which displayed high pseudotime values. We correlated these pseudotime findings with nuclei counts reported in the literature [37], allowing us to map each cluster to specific stages of germline development. This dual analysis of pseudotime distribution and developmental status percentage provided a comprehensive view of germline cell differentiation.

**Differential expression and gene ontology analysis.** To identify differentially expressed genes (DEGs) across various cell states and conditions, we used Seurat's FindMarkers function, employing a Wilcoxon rank-sum test with thresholds set for adjusted p-values below 0.05 and |log2 fold changes| greater than 0.5. Subsequent gene ontology analysis was performed using the ClusterProfiler package, which helped to understand the biological processes and pathways significantly associated with the identified DEGs. To visually compare gene expression across different samples and conditions, we used the DotPlot function within Seurat, facilitating an intuitive display of data that highlights specific gene expression changes.

**Availability of data.** The snRNA-seq data that support the findings of this study are publicly available from GEO repository with the identifier: GSE265917.

## Targeted metabolic analysis of nucleosides

**Sample collection.** Around 5000 *C. elegans* in each condition were collected in the bead beater tubes (Sarstedt Inc Screw Cap Micro tube 2ml) with 250 μL RNAlater Stabilization Solution (Invitrogen: AM7021) and 10μg/ml of tetrahydrouridine to prevent rapid deamination of the nucleosides [38] and then snap freezing in liquid nitrogen which were stored at -80°C until use.

**Nucleosides extraction and purification.** Nucleosides were extracted from the samples following a previously outlined method [39] with some modifications. Briefly, RNAlater Stabilization Solution was removed from thawed samples, and 250 μL LC-MS H2O with RNase Inhibitor and beads were added to the samples for homogenization with a bead beater. After homogenization, the sample was transferred to a 2 mL tube. pH adjustment to 8.5 was done using ammonium in methanol, followed by addition of methanol at a ratio of 1:4 (v:v). The mixture was then centrifuged at 14,000 rpm for 20 min at 4 °C after vortexing for 3 min. The supernatant was collected and dried using SpeedVac. Each dried sample was dissolved then in 250 μL water.

For nucleoside purification, each sample was loaded onto an OASIS HLB cartridge (Waters Corp., Milford, MA, USA) activated 3 times with water and methanol. After loading, cis-diol compounds were eluted with 250 μL 2.8% ammonium hydroxide (NH4OH) in methanol, repeated 3 times. Eluates were then dried using SpeedVac

Dried samples were reconstituted in 100 μL water, centrifuged at 14,000 rpm for 10 min at 4 °C, and the clear upper solution was transferred to an LC vial for LC-MS analysis.

**LC-MS/MS analysis.** The experiments were performed on an Orbitrap Fusion Lumos Tribrid Mass Spectrometer equipped with Ion Max API source housing with HESI-II probe and Vanquish UHPLC System (ThermoFisher Scientific). The separation was performed on an ACQUITY UPLC HSS T3 column (100 × 2.1 mm i.d., 1.8 μm) (Waters Corp., Milford, MA, USA) at a flow rate of 0.2 mL/min with column temperature of 40 °C (mobile phase A: 0.1% FA in water, mobile phase B: 0.1% FA in acetonitrile). The gradient was as follows: 0–3 min, 0-2.8% B; 3–9 min, 2.8%-10% B; 9-9.5 min, 10–30% B; 9.5-9.8 min, 30–60% B; 9.8-9.9, 60% B; 9.9-10 min, 60–0% B [39]. The targeted mode (parallel reaction monitor, PRM) was applied with Orbitrap MS acquired a full-scan survey in positive mode (m/z: 140–1300, automatic gain control target: standard, maximum injection time mode: auto, resolution at m/z 200: 120,000, the default charge state: 1, followed by tMS2 with precursor ions list of nitrogenous bases (mass range: normal, automatic gain control target: standard, maximum injection time mode: auto, Orbitrap with resolution at m/z 200: 60,000, collision-induced dissociation (CID) collision energy (%): 30 with activation time: 10 ms. Maximum injection time: 118 ms).

**LC-MS/MS data analysis.** The raw files were analyzed by Skyline [40–42]. The transition list was generated, applied and peak area was extracted and calculated by Skyline. All the samples were normalized based on protein concentrations. The sample concentration was calculated based on the standard curve.

**Quantification and statistical analysis.** Data were expressed as mean ± standard error of the mean (s.e.m.) and analyzed using GraphPad PRISM. Student's t-test (unpaired) compared the means of two groups. One-way ANOVA or two-way ANOVA followed by Holm–Sidak's or Holm-Bonferroni's corrections, as indicated in the figure legends, were applied. Statistical significance in figure legends is denoted by asterisks: ns (not significant, $p > 0.05$), *$p < 0.05$; **$p < 0.01$, ***$p < 0.001$, ****$p < 0.0001$. Detailed information on sample size, biological replicates, and statistical analysis for each experiment is provided in the figure legends. Figures and graphs were generated using BioRender, GraphPad Prism 10 (GraphPad Software), and Illustrator (CC 2019; Adobe). The researchers were not blinded during experiments or outcome assessments.

## Supporting information

**S1 Fig. ENT-1 and ENT-2 regulate reproduction independent of bacterial inputs.** (A) RNAi knockdown of *ent-1* on OP50 bacteria alters the daily progeny number compared to the EV control. $n = 17$(EV), n = 16(*ent-1* RNAi). (B) RNAi knockdown of *ent-2* on OP50 bacteria alters the daily progeny number compared to the EV control. $n = 17$(EV), n = 16(*ent-2* RNAi). (A) (C) Genome illustrations of *ent-1*$^{KO}$ and *ent-2*$^{KO}$ mutants. (D) *ent-1*$^{KO}$ mutants on OP50 bacteria show no significant alteration of daily progeny number or brood size compared to wild type (WT) worms. $n = 23$(N2), n = 22(*ent-1*$^{KO}$). (E) *ent-2*$^{KO}$ mutants on OP50 bacteria show no significant alteration of daily progeny number or brood size compared to WT. $n = 18$(N2), n = 18(*ent-2*$^{KO}$). (F) *ent-1*$^{KO}$ mutants on HT115 bacteria show no significant alteration of daily progeny number or brood size compared to WT. $n = 28$(N2), n = 27(*ent-1*$^{KO}$). (G) *ent-2*$^{KO}$ mutants on HT115 bacteria show no significant alteration of daily progeny number or brood size compared to WT. $n = 22$(N2), n = 22(*ent-2*$^{KO}$). (H) (I) RT-qPCR analysis shows that the *ent-2* mRNA level is upregulated in the *ent-1*$^{KO}$ mutant (H), and the *ent-1* mRNA level is elevated in *ent-2*$^{KO}$ (I). n = 6 biologically independent samples in each condition. (J) The mutants of *ent-3*, *ent-5*, *ent-6*, or *ent-7* display normal brood size compared to wildtype control, while *ent-4(ok2161)* has reduced brood size. Statistic: *$p<0.05$, **$p<0.01$, ***$p<0.001$, ****$p<0.0001$, ns $p>0.05$. Student's t-test (unpaired, two-tailed) was applied for the brood size graphs in A, B, D, E, F and G, and RT-qPCR results in H and I. Two-way ANOVA with Bonferroni's post hoc test for the daily progeny graphs in A, B, D, E, F and G. One-way ANOVA with Holm–Sidak correction for J. Data are shown as mean ± S.E.M. from at least three independent biological replicates, each with 5–10 animals, except (J) which is performed with two independent biological replicates, each with around 4animals. The "n" values represent the total number of animals across replicates.
(PDF)

**S2 Fig. Germline ENT-1 and intestinal ENT-2 cooperate in regulating reproduction.** (A) Bright-field images show sterile *ent-1*$^{KO}$; *ent-2*$^{KD}$ mutants and their controls. Scale bar: 100 μm. (B, C) Diagrams showing CRISPR knock-in strains: *ent-1::mNeonG*reen (B), *ent-2::wrmscarlet* (C). (D, E) Endogenous localization of ENT-1::mNeonGreen in the germline and intestine (D) and ENT-2::wrmscarlet in gonadal sheath cells and the intestine (E). Scale bar: 50 μm. (F) Images of *ent-1::mNeonGreen* in WT and *ent-2*$^{KO}$. Scale bar: 50 μm. (G) Images of *ent-2::wrmScarlet* in WT and *ent-1*$^{KO}$. Scale bar: 50 μm. (H) (I) Diagrams showing CRISPR knock-in strains: *degron::ent-1* (H) and *degron::ent-2* (I). (J) (K) Immunostaining results show the efficiency of tissue-specific degron-mediated protein degradation induced by the Auxin treatment in the *degron::ent-1* strain (J) and in the *degron::ent-2* strain (K). Scale bar: 50 μm.
(PDF)

**S3 Fig. Changes in the germline and other tissues in association with the loss of ENT-1 and ENT-2.** (A) Gene ontology enrichment analysis of biological processes was performed on the down-regulated expressed genes in the germ-line of *ent-1*$^{KO}$; *ent-2*$^{KD}$ worms compared to their controls, revealing pathways related to cell cycle and division are over-represented. (B) (C) Dotplots show the relative expression levels of genes that are associated with purine metabolism of control, *ent-1*$^{KO}$, *ent-2*$^{KO}$, and *ent-1*$^{KO}$; *ent-2*$^{KD}$ animals in the hypodermis (B) and muscle (C). (D) Schematic illustration of *C. elegans* homologous genes involved in pyrimidine metabolism pathways. (E) (F) Dot plots show the relative expression levels of genes associated with pyrimidine metabolism in the germline (E) and intestine (F).
(PDF)

## Acknowledgments

We thank A. Dervisefendic and P. Svay for their assistance with worm strain maintenance. We thank I. Neve for the great help in generating CRISPR knockout strains. Several strains were obtained from the *Caenorhabditis* Genetics Center (CGC), which is supported by the NIH Office of Research Infrastructure Programs (P40 OD010440).

## Author contributions

**Conceptualization:** Youchen Guan, Meng Wang.

**Data curation:** Youchen Guan.

**Formal analysis:** Youchen Guan.

**Funding acquisition:** Meng Wang.

**Investigation:** Youchen Guan, Yong Yu, Shihong Max Gao, Lang Ding, Qian Zhao.

**Methodology:** Youchen Guan, Meng Wang.

**Project administration:** Meng Wang.

**Supervision:** Meng Wang.

**Validation:** Youchen Guan.

**Writing – original draft:** Youchen Guan.

**Writing – review & editing:** Meng Wang.

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
