## [Decision Letter · Decision Letter 0]

Dear Dr Wang,

Thank you very much for submitting your Research Article entitled 'Cross-Tissue Coordination between SLC Nucleoside Transporters Regulates Reproduction in Caenorhabditis elegans' to PLOS Genetics.

The manuscript was fully evaluated at the editorial level and by independent peer reviewers. The reviewers appreciated the attention to an important problem, but raised some substantial concerns about the current manuscript. Based on the reviews, we will not be able to accept this version of the manuscript, but we would be willing to review a much-revised version. We cannot, of course, promise publication at that time.

If you decide to revise the manuscript for further consideration at PLOS Genetics, please aim to resubmit within the next 60 days, unless it will take extra time to address the concerns of the reviewers, in which case we would appreciate an expected resubmission date by email to plosgenetics@plos.org.

If present, accompanying reviewer attachments are included with this email; please notify the journal office if any appear to be missing. They will also be available for download from the link below. You can use this link to log into the system when you are ready to submit a revised version, having first consulted our Submission Checklist .

PLOS has incorporated Similarity Check , powered by iThenticate, into its journal-wide submission system in order to screen submitted content for originality before publication. Each PLOS journal undertakes screening on a proportion of submitted articles. You will be contacted if needed following the screening process.

To resubmit, log into your Editorial Manager account and select the option 'Revise Submission' in the 'Submissions Needing Revision' folder.

We are sorry that we cannot be more positive about your manuscript at this stage. Please do not hesitate to contact us if you have any concerns or questions.

Yours sincerely,

Sean P. Curran

Academic Editor

PLOS Genetics

Giovanni Bosco

Section Editor

PLOS Genetics

Reviewer's Responses to Questions

**Comments to the Authors:**

Reviewer #1: In this manuscript, Guan et al use sophisticated genetic methods to determine the tissue-specific requirements of the solute/nucleoside transporters ENT-1 and ENT-2 in the germline and intestine, respectively. They propose a model whereby nucleosides are release from the intestine via ENT-2 and subsequently taken up by the germ cells via ENT-1, and that this is required for proper reproductive health. Using snRNA-seq and metabolomics, they found that the guanosine was the primary metabolite that was exported by ENT-2 from the intestine and taken up by germ cells via ENT-1, and that this was essential for proper reproduction. The manuscript is very well-written, extremely easy to read, and the data is very strong. Overall, I believe the manuscript even in its current state is deserving of publication in PLoS Genetics and only have very minor comments, which I believe can strengthen the manuscript, but is certainly not essential.

-The authors argue that ent-1 and ent-2 loss of function (lof) have much milder phenotypes than a double lof due to the compensation by the upregulation of the other transporter. However, this is inconsistent with a model whereby ENT-1 functions solely in the germline and ENT-2 functions solely in the intestine. Is it possible that ENT-1 primarily functions in the germline, but has the capacity to also function as a nucleoside exporter in the intestine when ENT-2 is lost? Similar for ENT-1? Since the authors have made reporters for both lines, it might be useful to see whether the increase in ENT-1 upon ent-2 lof occurs in both the germline and intestine, and the same for ENT-2 upon ent-1 lof. Minimally, the authors can analyze the snRNAseq data for this information and then speculate on something in the discussion about this conundrum.

-The discussion seems very short. I feel several things can be expanded upon. For example, why do the authors believe that the other ENTs are not required for reproduction? Why do the authors believe that ENT-1 and ENT-2 are not required for development, despite nucleosides regulation being very important for cell division as they say in their introduction? How does the work fit in with other models of reproduction? Are these transporters essential in higher eukaryotic systems as well?

-Line 214: “conduced” should be “conducted”?

Reviewer #2: Here, Guan, et al. report that they found that knockdown of ent-1 or ent-1 slight reduces total progeny number, and the double reduction has a larger effect on reproduction. GFP tagging revealed tissue specificity of ENT-1 and -2, in germline and intestine vs gonad and intestine, respectively. Degradation of the ENTs in specific tissues suggested that ENT-1 in germline and ENT-2 in intestine function to regulate reproduction. Single-nuc seq analysis suggests that the double mutants have fewer proliferating germ cells, and that (surprisingly?) purine metabolism genes are upregulated in the germline. LC/MS showed that there are differences in levels of G, A, and I , but not C, between controls and the double mutants.

Microinjection of G into the pseudocoelom at L4 increased brood size, but A did not.

1. What is the percent effect of progeny number reduction? It looks extremely slight (Fig. 1C, 1D). Please report this % in the text and in the figure legend.

2. The terminology isn’t that it “increased” progeny in the later reproduction – this could just be slowing of egg-laying.

3. (Out of curiosity - How does one propagate a CRISPR strain with no progeny production? )

4. I understand that the authors have previously published on single-nucleus sequencing, but why is that a relevant experiment here, if the authors already showed where the proteins function and that the relevant changes are in the germline to regulate reproduction? (How would this be different than whole-worm analyses? Were there actually differences between single cells that would have been invisible in whole-worm analyses?) The logic is unclear.

5. What would the single mutant snSeq show in specific tissues? This would go far to support the hypothesis they later state about the intestinal role

6. The pseudo-time analysis is nice and should be in the main figures.

7. It looks like there are two distinct populations in the guanosine microinjections – do the authors know what might have led to this huge difference in brood size (<50 vs >200)?

8. In Figure 4F, how did the authors come to their conclusions in this table? Are these data? If so, from what experiment did these significance (or whatever the +++ means) come from? Or is this a model? If the latter, it needs to be much clearer in the text, and perhaps depicted as a model rather than as if it is showing data (but if the latter, the data need to be clearer).

How does the current work fit into the context of previously known work from Min Han’s lab on the role of nucleotides and nucleotide metabolism and nucleotide-sensing mechanisms for controlling reproduction (Jia et al. 2020; Chi et al. 2016)?

Minor:

- Change “Our recent findings found” to “We recently found”

- seems excessive to claim this as a “pioneering” example

- not “in human” but “in humans”

Reviewer #3: This manuscript by Guan and colleagues details the intricate trafficking of nucleosides and nucleotides between somatic and germline tissues via the SLC29A family of transporters using C. elegans as a system. The authors identify a pair of equilibrative nucleoside transporters ENT-2 and ENT-1 which act in soma and germline respectively to fulfill organismal nucleoside needs for reproduction. Loss of ent-1 and ent-2 prompts synthetic lethality, indicating cooperative roles in nucleoside and nucleotide homeostasis in the intact organism. The authors go on to use elegant and overlapping lines of evidence to uncover tissue-specific roles of each transporter. The use of snRNAseq illuminates tissue specific transcriptional responses to the loss of either or both ENT genes, and metabolomics subsequently elucidates precisely the nucleosdies altered with ENT loss. The methods used are rigorous and results clearly presented. I do have some concerns with the statistics used and failure to clearly indicate numbers of biological replicates, however I remain enthusiastic overall for the novelty and findings presented by the authors herein. Also, some alternative possibilities exist for why there is synthetic lethality with dual ent1/2 loss, and these should be explored and/or discussed.

MAJOR:

Are deletion mutants available for the ent genes, and if so, are they viable? Over-reliance on RNAi in the initial ENT gene survey might mask deleterious effects of loss of ent genes other than ent-1 and ent-2.

This reader sincerely appreciates the inclusion of detailed statistical information on sample size in each figure legend. None-the-less, several of the statistics are improperly applied (e.g. t-tests without multiple hypothesis testing as in multiple comparisons in Fig. 1E), and it remains unclear from the data presented how many times the experiment was repeated, and whether these “n” are from one biological replicate or whether the experiment was sufficiently replicated.

T-tests were again inappropriately applied when two-way ANOVA or similar with post-hoc correction should have been used for Figures 2C, D, E and one-way ANOVA or similar for figures 4G-H.

A possibility remains that the synthetic lethality in double ent-1 and ent-2 KO is because of combined action in germline (or intestine). While this is unlikely, the authors have the ability to test combined AID of both ENT-1 and ENT-2 in germline and intestine. This dual loss would support their model quite convincingly beyond the add-back and single conditional AID experiments. Also it could explain why there is no visible drop in fecundity in spite of reduced guanosine levels (Fig. 4) if the more severe drop with double loss was exhibited in specific tissues which are not resolvable by whole animal LC-MS. If the authors do not think this strengthens their story they should at least indicate why they believe this experiment to be unnecessary.

Why are Adenosine and inosine rising so dramatically in the single ent and double ent loss? Is the drop in gmps-1 expression in the salvage pathway partly to blame? The authors should explain why they are seeing these dramatic shifts away from G towards A/I.

While I think the model is compelling, alternative explanations exist, and unless the authors invoke non-transporter mediated G uptake in germline in the microinjection experiments in 4G, we can surmise that additional SLCs participate in uptake of G into germline or compensation for loss of ent-1ko. Could dual ent-1 and ent-2 roles in the germline be an alternate explanation? There are other possibilities as well which the authors should at least briefly discuss and/or test if easy to do so.

MINOR:

There are a few minor grammatical errors, e.g. ln 34 “collectively termed as metabolites.” Should probably be “collectively termed metabolites…”

Ln133 “exon sequences” should probably specify “coding exonic nucleotide sequence” or similar.

“Offsprings” in Y-axis labels should probably be “offspring”

It appears that brood is already reduced modestly in the -Auxin AID strains, suggestive of either a negative pleiotropy of tagging or that there is modest constitutive reduction in ENT expression. Either way these possibilities are informative for the field and should at least be mentioned in the manuscript for those less experienced in making use of the AID system in vivo.

**Have all data underlying the figures and results presented in the manuscript been provided?**

Reviewer #1: Yes

Reviewer #2: Yes

Reviewer #3: Yes

PLOS authors have the option to publish the peer review history of their article (what does this mean? ). If published, this will include your full peer review and any attached files.

**Do you want your identity to be public for this peer review?** For information about this choice, including consent withdrawal, please see our Privacy Policy .

Reviewer #1: **Yes: ** Ryo Higuchi-Sanabria

Reviewer #2: No

Reviewer #3: No

---

## [Decision Letter · Decision Letter 1]

PGENETICS-D-24-01044R1

Cross-Tissue Coordination between SLC Nucleoside Transporters Regulates Reproduction in Caenorhabditis elegans

PLOS Genetics

Dear Dr. Wang,

Thank you for submitting your manuscript to PLOS Genetics. After careful consideration, there is only one minor concern raised by the reviewers. Therefore, we invite you to submit a revised version of the manuscript that addresses this singular point prior to acceptance.

Please submit your revised manuscript within 30 days Mar 13 2025 11:59PM. If you will need more time than this to complete your revisions, please reply to this message or contact the journal office at plosgenetics@plos.org. Please include the following items when submitting your revised manuscript:

We look forward to receiving your revised manuscript.

Kind regards,

Sean P. Curran

Academic Editor

PLOS Genetics

Giovanni Bosco

Section Editor

PLOS Genetics

Aimée Dudley

Editor-in-Chief

PLOS Genetics

Anne Goriely

Editor-in-Chief

PLOS Genetics

**Additional Editor Comments :**

Please see the comment from reviewer 2, which hopefully can be easily edited to address the concern.

**Journal Requirements:**

At this stage, the following Authors/Authors require contributions: Lang Ding. Please ensure that the full contributions of each author are acknowledged in the "Add/Edit/Remove Authors" section of our submission form.

The list of CRediT author contributions may be found here: https://journals.plos.org/plosgenetics/s/authorship#loc-author-contributions

2) Please include the affiliation of Meng Wang in the online submission form.

**Reviewers' comments:**

Reviewer's Responses to Questions

Reviewer #1: The authors have very comprehensively addressed all reviewer concerns. I highly recommend the manuscript for publication.

Reviewer #2: While the authors have addressed most of my comments, the table in Figure 4F continues to be misleading and should be removed entirely, as it is too easily misconstrued as data when in fact it is not. I realize that the authors have added "Predicted [sic]" to the top of the table, but again, it is not appropriate to depict this model as a table. The authors should figure out how to present their model in a way that does not suggest that they have actually made these measurements of guanosine in tissues, which they have not done - it is too easy for readers to mistake this table as data, and believe that the authors have somehow developed tissue-specific measurements of nucleotides. Instead, they should make a graphical figure of their predictions. I'm sorry that this is extra work, but it is dangerous to present models in formats normally used for data.

Reviewer #3: The authors have answered all of my concerns. The mansucript is well conducted and rigorously supported.

**Have all data underlying the figures and results presented in the manuscript been provided?**

Reviewer #1: Yes

Reviewer #2: Yes

Reviewer #3: Yes

PLOS authors have the option to publish the peer review history of their article (what does this mean? ). If published, this will include your full peer review and any attached files.

**Do you want your identity to be public for this peer review?** For information about this choice, including consent withdrawal, please see our Privacy Policy .

Reviewer #1: **Yes: ** Ryo Higuchi-Sanabria

Reviewer #2: No

Reviewer #3: No

**Figure resubmission:**
---

## [Editor Report · Decision Letter 2]

Dear Dr Wang,

Thank you for sending your revised manuscript. We are pleased to inform you that your manuscript entitled "Cross-Tissue Coordination between SLC Nucleoside Transporters Regulates Reproduction in Caenorhabditis elegans" has been editorially accepted for publication in PLOS Genetics. Congratulations!

Yours sincerely,

Anne Goriely

Editor-in-Chief

PLOS Genetics

Aimée Dudley

Editor-in-Chief

PLOS Genetics

Comments from the reviewers (if applicable):

**Data Deposition**

http://datadryad.org/submit?journalID=pgenetics&manu=PGENETICS-D-24-01044R2

**Press Queries**

---

## [Editor Report · Acceptance letter]

PGENETICS-D-24-01044R2

Cross-Tissue Coordination between SLC Nucleoside Transporters Regulates Reproduction in Caenorhabditis elegans

Dear Dr Wang,

We are pleased to inform you that your manuscript entitled "Cross-Tissue Coordination between SLC Nucleoside Transporters Regulates Reproduction in Caenorhabditis elegans" has been formally accepted for publication in PLOS Genetics! Your manuscript is now with our production department and you will be notified of the publication date in due course.

With kind regards,

Zsofia Freund

PLOS Genetics

On behalf of:
